# Quality of life among patients with atrial fibrillation: A theoretically-guided cross-sectional study

Kathy L. Rush[1]*, Cherisse L. Seaton[1], Lindsay Burton[1], Peter Loewen[2], Brian P. O'Connor[3], Lana Moroz[4], Kendra Corman[1], Mindy A. Smith[5], Jason G. Andrade[4,6]

1 School of Nursing, University of British Columbia, Okanagan, Kelowna, BC, Canada, 2 Faculty of Pharmaceutical Sciences, University of British Columbia, Vancouver, Vancouver, BC, Canada, 3 Department of Psychology, University of British Columbia, Okanagan, Kelowna, Canada, 4 Cardiac Atrial Fibrillation Specialty Clinic, Vancouver General Hospital, Vancouver, BC, Canada, 5 Department of Family Medicine, Michigan State University, East Lansing, Michigan, United States of America, 6 Department of Medicine, University of British Columbia, Vancouver, Canada

* kathy.rush@ubc.ca

## Abstract

### Background

Patients with atrial fibrillation (AF) have significantly lower health-related quality of life (HRQoL) compared to the general population and patients with other heart diseases. The research emphasis on the influence of AF symptoms on HRQoL overshadows the role of individual characteristics. To address this gap, this study's purpose was to test an incremental predictive model for AF-related HRQoL following an adapted HRQoL conceptual model that incorporates both symptoms and individual characteristics.

### Methods

Patients attending an AF specialty clinic were invited to complete an online survey. Hierarchical regression analyses were conducted to examine whether individual characteristics (overall mental health, perceived stress, sex, age, AF knowledge, household and recreational physical activity) incremented prediction of HRQoL and AF treatment satisfaction beyond AF symptom recency and overall health.

### Results

Of 196 participants (mean age 65.3 years), 63% were male and 90% were Caucasian. Most reported 'excellent' or 'good' overall and mental health, had high overall AF knowledge scores, had low perceived stress scores, and had high household and recreation physical activity. The mean overall AF Effect On Quality-Of-Life Questionnaire (AFEQT) and AF treatment satisfaction scores were 70.62 and 73.84, respectively. Recency of AF symptoms and overall health accounted for 29.6% of the variance in overall HRQoL and 20.2% of the variance in AF treatment satisfaction. Individual characteristics explained an additional 13.6% of the variance in overall HRQoL and 7.6% of the variance in AF treatment satisfaction. Perceived stress and household physical activity were the largest contributors to

**Data Availability Statement:** The dataset has been uploaded to Dryad and has been assigned the following unique digital object identifier (DOI): doi:10.5061/dryad.gtht76hsf.

**Funding:** This work was supported by a Canadian Institutes of Health Research (https://cihr-irsc.gc.ca) Project Grant, award number PJT-148737; principal investigator, KLR. The funders had no role in study design, data collection and analysis, decision to publish, or preparation of the manuscript.

**Competing interests:** I have read the journal's policy and the authors of this manuscript have the following competing interests: JGA reports grants from Medtronic and the Heart and Stroke Foundation of Canada during the conduct of this study; personal fees from Medtronic and Biosense Webster Inc, outside the submitted study. This does not alter our adherence to PLOS ONE policies on sharing data and materials. The other authors have no conflicts to declare.

overall HRQoL, whereas age and AF knowledge made significant contributions to AF treatment satisfaction.

## Conclusions

Along with AF symptoms and overall health, individual characteristics are important predictors of HRQoL and AF treatment satisfaction in AF patients. In particular, perceived stress and household physical activity could further be targeted as potential areas to improve HRQoL.

## Introduction

Atrial fibrillation (AF) is a cardiac arrhythmia affecting 3% of the population worldwide [1], and frequently impairs patients' quality of life [2]. Compared to the general population and patients with other heart diseases (e.g., coronary heart disease), patients with AF have significantly lower health-related quality of life (HRQoL) [3–5]. Efforts to assist patients in managing AF, requires understanding of the aspects of their illness that present the greatest challenges and opportunities to improve their HRQoL, including their general health and well-being, treatment concerns, and satisfaction with their AF treatment.

A wealth of research has established that AF symptoms are related to HRQoL among patients with AF [6]. Emphasis on the influence of AF symptoms on HRQoL has not often included the role of individual characteristics such as demographic, psychological, and behavioral attributes of patients. Yet these characteristics have factored prominently in established conceptual models of HRQoL [7, 8]. Recent empirical evidence has begun to show the contribution of these individual characteristics to HRQoL in patients with AF. For example, anxiety has been found to be one of the factors most consistently associated with HRQoL [6].

Despite this beginning work, there remain a number of under-explored individual characteristics that have been linked to HRQoL in patients with AF, including perceived stress, physical activity/exercise, and AF knowledge. For example, research has reported that perceived stress contributes to mental health and anxiety among patients with AF [9] and perceived stress in conjunction with AF symptoms can be a primary trigger or cause of AF symptom episodes and influence HRQoL [10–13]. Additionally, physical activity/exercise has been suggested to play a role in reducing AF burden and improving AF-related symptoms and QoL [14, 15]. Indeed, following cardioversion, exercise was positively related to QoL among patients with AF that returned to and maintained sinus rhythm [16]. Increasing patients' AF knowledge may be an effective means of promoting symptom management and potentially improving HRQoL [17]. Not only have these relevant characteristics been under-studied but also the extent of their contributions relative to symptoms remains unknown. Therefore, the purpose of this study was to examine the extent to which individual characteristics (age, sex, mental health, stress, AF knowledge, and physical activity) contribute to AF HRQoL and treatment satisfaction while controlling for AF symptoms and overall health.

## Material and methods

### Study design and setting

We used a cross-sectional design that was guided by an expansion of Ferrans et al.'s [8] revised version of Wilson and Cleary's HRQoL model [7]. This involved including individual

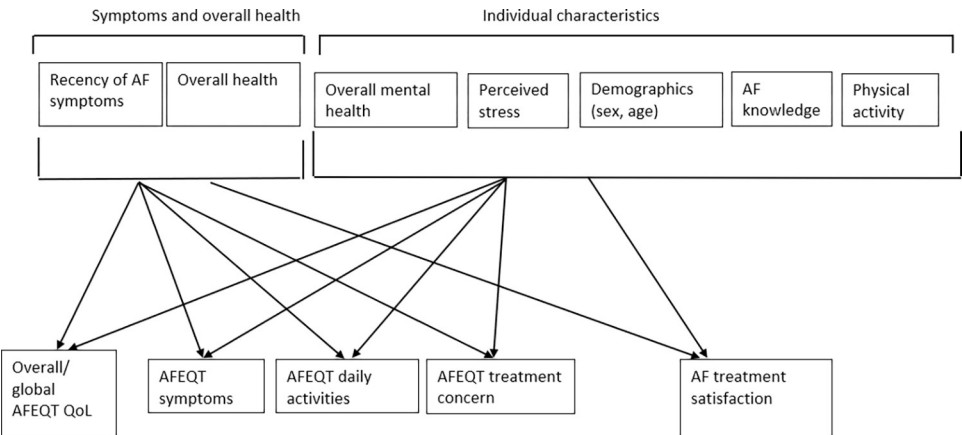

**Fig 1. Theoretical model for prediction of Quality-of-Life measures based on patient characteristics.** Note: model adapted from Ferrans et al. [8] model based on Wilson and Clearly [7].

characteristics of AF patients based on current evidence [6]. The model is displayed in Fig 1. The study was carried out in collaboration with a specialized AF clinic in an urban Western Canada area. Upon referral, the clinic provided integrated AF specialty care including acute interventions, education, disease management, and advanced treatments (e.g., ablation). The multi-disciplinary AF care team included cardiologists, electrophysiologists, nurse practitioners, pharmacists, and registered nurses. Since the onset of the COVID-19 pandemic clinic appointments have primarily been conducted remotely by telephone. University Behavioural Research Ethics Board approval was obtained [REB # anonymous for review]. The conduct and reporting of the study followed the Strengthening the Reporting of Observational Studies in Epidemiology (STROBE) Statement for reporting cross-sectional studies [18].

## Sample and recruitment

All patients of the clinic with an AF diagnosis who were over 18 years and could complete an online survey or had a family member who could assist, were eligible to participate. The clinic's booking clerk sent a letter detailing the research study (by mail or email) to all patients with upcoming appointments during the recruitment period. The letter informed patients of the ongoing study and to expect a telephone initiation from a research team member regarding their eligibility and interest in the study. Patient contact information was then shared with the research team using secure file transfer. Subsequently a research assistant (a physician or a licensed practical nurse) who had no prior relationship with participants contacted patients by telephone. Recruitment began in November 2020 and continued for one year until a sample size of approximately 200 was achieved. A post hoc power analysis assuming a medium effect size estimated required sample size for modelling to be 114, indicating appropriate sample size had been achieved for analyses [19].

## Data collection

Study data were collected using an online survey hosted on Qualtrics (Qualtrics, Provo, UT). Prior to taking the survey, all participants gave electronic consent. Participants who finished the survey were eligible for a chance to win one of three $150 gift certificates through a random draw. Clinical participant data were obtained via clinician referral letters as part of another project [20]. Clinical patient history data included treatment history, chronic disease history,

number of chronic conditions, medication, and cardiac risk factors. Type of AF (paroxysmal, persistent, permanent), and reason for referral (e.g., ablation consultation), were also included if reported by the referring clinician. Participants were given a unique survey ID and all identifying information was removed prior to data analysis.

## Measures

**Recency of AF symptoms.** Participants were asked "When was the last time you were aware of having an episode of atrial fibrillation?" Ordinal response choices were coded as (1) currently in AF or earlier today, (2) within the past week, (3) within the past month, (4) 1 month to 1 year ago, (5) more than 1 year ago, and (6) I was never aware of having atrial fibrillation; an interval based on the AFEQT that acts somewhat as an internal control [21]. Additionally, these choices were collapsed into another binary variable "AF symptoms < or > 1 month" representing if the patient had experienced symptoms over one month ago or within the past month. Patients who identified as asymptomatic were included in the "over one month ago" group.

**Overall health.** Participants were asked to rate their overall health on a scale ranging from 1 (poor) to 4 (excellent) [22].

**Overall mental health.** Participants were asked to rate their overall mental health on a scale ranging from 1 (poor) to 4 (excellent) [23].

**Perceived stress [24].** The Perceived Stress Scale (PSS-10), a 10-item, 5-point scale, measures the degree to which situations in one's life are appraised as stressful, ability to control aspects of life, confidence in handling problems, or being unable to cope with demands. The PSS-10 previously had a reliability alpha of .78 and correlated in a predictable way with other measures of stress [24].

**Socio-demographic characteristics.** These included sex, age, marital status, race/ethnicity, education, and income.

**AF knowledge [25].** The Knowledge about AF tool is a 28-item multiple choice-style questionnaire including questions about AF symptoms, treatment, medications, risk factors, and lifestyle. Participants are asked to choose one of 3 options for each question, only one of which is the correct response. The tool was developed using research on gaps in patient knowledge and patient values and management preferences. Knowledge scores are calculated as a percentage of correct answers, with higher numbers indicating higher knowledge. Four items were removed from the overall knowledge percent scores, as per McCabe et al. [25] finding that these items had factor loadings below .45 and were not reliable predictors of overall test performance and knowledge. The Knowledge about AF tool demonstrated an internal consistency reliability coefficient of .86, convergent validity (i.e., was positively related to another knowledge about AF test), and distinguished between patients recently diagnosed with AF from those seeking advanced treatment for AF [25].

**Household and recreational physical activity [26].** Physical activity scores are calculated based on the Phone-FITT questionnaire. The questionnaire was originally designed to be administered via telephone, so it was adapted for the use in an online survey by displaying the questions as checkboxes. First, participants were asked to indicate their participation in various household and recreational activities in a typical week within the past month, or specific months for season dependent activities. If they participated in an activity, participants were also asked to provide a frequency (times per week, and months per year for seasonal activities only), as well as choose a duration from 1 (1–15 minutes) to 4 (1 hour or more) and an intensity from 1 (breathing normally and able to carry on a conversation) to 3 (too out of breath to carry on a conversation). Scores are calculated as the sum of the frequency, duration, and

intensity for all household and all recreational activities, with higher scores indicating higher physical activity. The Phone-FITT has previously demonstrated test-retest reliability as well as convergent validity (i.e., positive correlations with accelerometer counts) [26].

**Atrial Fibrillation Effect on QualiTy-of-Life Questionnaire (AFEQT) [21].** The AFEQT is a 20-item, 7-point scale comprising overall HRQoL and three sub-domains: symptoms, daily activities, treatment concerns, along with AF treatment satisfaction. The AFEQT was developed for use as an outcome measure in trials and interventions and for disease management. In the present study, Cronbach α reliability coefficient was >0.88 for the AFEQT overall score and the symptoms (0.95), daily activities (0.94), treatment concern (0.90), and .88 for treatment satisfaction. Overall HRQoL scores are calculated as the sum of items 1–18, accounting for unanswered items, and normed on a scale from 0–100, with higher numbers indicating higher HRQoL. Treatment satisfaction scores follows the same calculation for items 19–20, with higher scores indicating higher treatment satisfaction; while it is included in the AFEQT, AF treatment satisfaction is not calculated in the overall HRQoL with the other sub-scales. A difference of + or minus 5 points on the AFEQT are clinically meaningful [2].

## Data cleaning

Two-hundred and three participants responded to the survey. Seven participants were missing one third or more of the scale scores and were removed. Of the remaining 196 participants, less than 1% of data were missing for variables included in the primary analyses. Missing data were replaced using multiple imputation [27] A large portion of clinical data obtained from referrals was missing, and ranged from 24.0% (use of anticoagulants) to 50.5% (type of AF). Missing referral data were not replaced.

## Analysis

Descriptive statistics were used to summarize patient characteristics and socio-demographic data. We conducted bivariate analyses using Wilcoxon rank sum test, Pearson's Chi-squared test, and Fisher's exact test to evaluate relationships between the dichotomized AF symptoms < or > 1 month with all variables because experiencing AF symptoms within the past 4 weeks is a clinically meaningful cutoff, and the AFEQT scores specifically assess symptoms "over the past 4 weeks" [21].

Spearman's rank order (Rho) correlations were conducted to examine preliminary associations between all study variables. In addition, multidimensional scaling analyses were conducted to examine the associations between all variables visually and guided our decisions when to enter each of the predictors in the regression models. Five separate linear regressions were conducted with overall HRQoL, along with the component AFEQT sub-scales and AF treatment satisfaction as the dependent variables. Recency of AF symptoms and overall health were entered as predictors on step 1, overall mental health and perceived stress as predictors of step 2, and sex, age, AF knowledge, and household and recreational physical activity as predictor variables on step 3. Cohen's [28] benchmarks (i.e., 2% = small, 15% = medium, and 35% = large) were used to interpret strength of the variance accounted for by variables in each step of the regressions. P-values less than 0.05 were considered statistically significant.

Standardized statistical diagnostics for determining normality, detecting outliers, and ensuring that the data met the assumptions of regression were performed. Where necessary, variables were Windsorized to improve normality of distributions. All regression analyses met assumptions of linearity, heteroscedasticity, and multicollinearity. Analyses were performed using R [29] and IBM[TM] SPSS software (version 28) [30].

## Results

### Characteristics of the study population

Of the 579 patients eligible for inclusion, 352 (61%) agreed to be sent the online survey invitation. Of those, 203 started the survey and 196 completed the survey (response rate = 56%). Participants were an average age of 65.28 years (range 33 to 91 years, SD = 10.26), primarily male (n = 123, 63%), and Caucasian (n = 176, 90%). Characteristics of the study populations are shown in Table 1.

Despite over half of patients (59%) experiencing AF symptoms within the past month, the majority of participants had 'excellent' or 'good' overall health (73%) and mental health (89%) ratings. Overall knowledge scores were high (83%), ranging from 29% to 100% and with highest scores for basic AF knowledge (97%), and lowest scores for knowledge of the consequences of untreated AF (65%). On average perceived stress scores were low according to scale guidelines [24]. Household and recreation physical activity were on average high, compared to a sample of adults aged 70 years or older enrolled in an adult exercise program [26]. The mean overall AFEQT score in our sample was 70.6, with a range of 7.4 to 100. The mean overall AF treatment satisfaction score was 73.8, with a range of 0 to 100. As shown in Table 1, those experiencing AF symptoms within the past month had significantly lower ratings for overall health, HRQoL, treatment satisfaction, and higher perceived stress.

In terms of clinical descriptors, referring clinicians stated that 23 (11.7%) of patients were newly diagnosed and 85 (43%) referrals mentioned ablation consultation. Patients had on average 2.45 co-morbidities and 5% listed congestive heart failure. Of those recorded, patient participants' AF type was paroxysmal (69%), persistent (27%), or permanent (2%). Approximately 46% had undergone previous cardioversion, 26% had undergone previous ablation, and 3% had pacemaker implantations. The majority (62%) were prescribed anticoagulants and patients were taking an average of 2.87 medications.

### Model building

Correlations between all continuous (and ordinal) variables are presented in Table 2. Initial inspection of these bivariate correlations suggested the importance of recency of AF symptoms, overall health, overall mental health and perceived stress for HRQoL and AF treatment satisfaction. A multidimensional scaling analysis with all variables entered simultaneously (i.e., weighted equally) further confirmed our hypothesis that recency of AF symptoms and overall health were the most closely related to HRQoL (i.e., should be entered first in the regression models), followed by mental health variables (overall mental health and perceived stress), with all other variables more distally related to HRQoL and AF treatment satisfaction.

### Regressions

See Table 3 for the results of regression analyses examining the association between predictors (recency of AF symptoms, overall health, overall mental health, perceived stress, sex, age, AF knowledge, household PA, and recreational PA) and overall HRQoL, three HRQoL subscales, and treatment satisfaction (outcomes).

### HRQoL and subscales

On Step 1 of the regressions, length of time since participants last had AF symptoms was positively related to all three HRQoL subscales and individuals' overall HRQoL. In addition, higher overall health was positively related to the daily activities HRQoL subscale, and overall HRQoL. These two variables together contributed 29.6% of the variance in overall HRQoL,

**Table 1. Participant characteristics.**

| Characteristics | All Participants (*n* = 196) | AF Symptoms < or > 1 month | | *p*-value |
| | | Within the past month (*n* = 115) | Over a month ago (*n* = 81) | |
|---|---|---|---|---|
| **Age[1]** | 65 (10) | 65 (10) | 66 (11) | 0.56[3] |
| **Sex[2]** | | | | 0.34[4] |
| Female | 73 (37) | 46 (40) | 27 (33) | |
| Male | 123 (63) | 69 (60) | 54 (67) | |
| **Ethnicity[2]** | | | | 0.21[5] |
| Caucasian/White | 176 (91) | 103 (91) | 73 (91) | |
| Asian | 15 (7.8) | 10 (8.8) | 5 (6.2) | |
| Indigenous | 2 (1.0) | 0 (0) | 2 (2.5) | |
| Missing | 3 | 2 | 1 | |
| **Marital Status[2]** | | | | 0.71[5] |
| Single, divorced, separated, or widowed | 47 (24) | 27 (23) | 20 (25) | |
| Married, remarried or common law | 147 (75) | 86 (75) | 61 (75) | |
| Missing | 2 | 2 | 0 | |
| **Education[2]** | | | | 0.29[4] |
| College, University, Graduate or Professional Degree | 129 (66) | 71 (62) | 58 (72) | |
| Some post-secondary | 37 (19) | 21 (18) | 14 (17) | |
| High School or less | 30 (15) | 21 (18) | 9 (11) | |
| **Income[2]** | | | | 0.29[5] |
| Less than $25,000 | 13 (6.6) | 10 (8.7) | 3 (3.7) | |
| $25,000-$50,000 | 38 (19) | 25 (22) | 13 (16) | |
| $51,000-$75,000 | 41 (21) | 26 (23) | 15 (19) | |
| Over $75,000 | 99 (51) | 51 (44) | 48 (59) | |
| Missing | 5 | 3 | 2 | |
| **Overall Health[2]** | | | | 0.002[5] |
| Excellent | 26 (13) | 10 (8.7) | 16 (20) | |
| Good | 117 (60) | 63 (55) | 54 (67) | |
| Fair | 44 (22) | 35 (30) | 9 (11) | |
| Poor | 6 (3.1) | 4 (3.5) | 2 (2.5) | |
| **Overall Mental Health[2]** | | | | 0.29[5] |
| Excellent | 74 (38) | 41 (36) | 33 (41) | |
| Good | 99 (51) | 58 (50) | 41 (51) | |
| Fair | 22 (11) | 16 (14) | 6 (7.4) | |
| Poor | 1 (0.5) | 0 (0) | 1 (1.2) | |
| **Perceived Stress[1]** | 12 (6) | 13 (7) | 11 (6) | 0.009[3] |
| Missing | 5 | 1 | 4 | |
| **AF Knowledge (Overall)[1]** | 83 (12) | 84 (12) | 82 (13) | 0.32[3] |
| Basic AF Knowledge | 97 (12) | 98 (10) | 96 (15) | |
| Common symptom knowledge | 94 (16) | 95 (14) | 92 (19) | |
| Consequences knowledge | 65 (21) | 66 (23) | 63 (18) | |
| Recurrent knowledge | 82 (23) | 83 (22) | 81 (25) | |
| Treatment Knowledge | 78 (21) | 79 (21) | 76 (21) | |
| Monitoring knowledge | 86 (15) | 88 (14) | 84 (16) | |
| Risk factors knowledge | 93 (16) | 93 (16) | 92 (18) | |
| Psyc knowledge | 100 (0) | 100 (0) | 100 (0) | |

(*Continued*)

**Table 1.** (Continued)

| Characteristics | All Participants (n = 196) | AF Symptoms < or > 1 month | | p-value |
| --- | --- | --- | --- | --- |
| | | Within the past month (n = 115) | Over a month ago (n = 81) | |
| Missing (overall) | 4 | 2 | 2 | |
| **Household Activity[1]** | 32 (13) | 31 (14) | 33 (11) | 0.19[3] |
| **Recreational Activity[1]** | 37 (21) | 37 (22) | 36 (20) | 0.74[3] |
| **Overall HRQoL (AFEQT)[1]** | 71 (21) | 61 (20) | 84 (15) | < .001[3] |
| Symptoms Subscale | 75 (21) | 65 (21) | 89 (12) | < .001[3] |
| Daily Activities Subscale | 70 (27) | 60 (26) | 84 (22) | < .001[3] |
| Treatment Concern | 68 (22) | 61 (23) | 80 (16) | < .001[3] |
| **Treatment Satisfaction (AFEQT)[1]** | 74 (26) | 66 (26) | 86 (22) | < .001[3] |
| Missing | 2 | 0 | 2 | |

[1] Mean (SD)

[2] n (%)

[3] Wilcoxon rank sum test

[4] Pearson's Chi-squared test

[5] Fisher's exact test

**Table 2. Correlations between all study variables.**

| | 1 | 2 | 3 | 4 | 5 | 6 | 7 | 8 | 9 | 10 |
| --- | --- | --- | --- | --- | --- | --- | --- | --- | --- | --- |
| | r (p) | r (p) | r (p) | r (p) | r (p) | r (p) | r (p) | r (p) | r (p) | r (p) |
| 1. Overall HRQoL | 1.0 | | | | | | | | | |
| 2. AF Treatment Satisfaction | .511 (< .001) | 1.0 | | | | | | | | |
| 3. Recency of AF Symptoms | .475 (< .001) | .459 (< .001) | 1.0 | | | | | | | |
| 4. Overall Health | .431 (< .001) | .329 (< .001) | .278 (< .001) | 1.0 | | | | | | |
| 5. Overall mental health | .189 (.008) | .247 (< .001) | .046 (.526) | .397 (< .001) | 1.0 | | | | | |
| 6. Perceived stress | -.385 (< .001) | -.221 (.002) | -.122 (.088) | -.248 (< .001) | -.515 (< .001) | 1.0 | | | | |
| 7. Age | -.082 (.253) | .123 (.086) | .096 (.180) | -.015 (.830) | .174 .014) | -.115 (.108) | 1.0 | | | |
| 8. AF knowledge | -.005 (.944) | -.153 (.032) | -.117 (.103) | .041 (.564) | .030 (.672) | -.020 (.783) | .057 (.430) | 1.0 | | |
| 9. Household activity | .155 (.030) | .091 (.207) | .082 (252) | .058 (.423) | -.020 (.785) | .022 (.764) | -.104 (.148) | .033 (.651) | 1.0 | |
| 10. Recreational activity | .051 (.482) | .065 (.366) | .015 (.830) | .204 (.004) | .005 (.942) | -.103 (.149) | .082 (.253) | .266 (< .001) | .301 (< .001) | 1.0 |

Note: Correlations based on Spearman's Rho; HRQoL = Health-Related Quality of Life

**Table 3. Regression analyses examining the association between recency of AF symptoms and overall health along with individual characteristics and HRQoL outcomes and treatment satisfaction.**

| | B (95% CI) | β | Coefficient P value | Partial r | Semi-partial r | R Square | Model F (df) | F change P value |
|---|---|---|---|---|---|---|---|---|
| **Overall HRQoL** | | | | | | | | |
| Step 1 | | | | | | .296 | 40.49 (2, 195) | < .001 |
| Recency of AF symptoms | 5.32 (3.58, 7.17) | .38 | < .001 | .40 | .36 | | | |
| Overall health | 9.73 (5.50, 13.64) | .31 | < .001 | .34 | .30 | | | |
| Step 2 | | | | | | .386 | 30.07 (4, 195) | < .001 |
| Overall mental health | -3.97 (-8.51, .603) | -.12 | .089 | -.12 | -.10 | | | |
| Perceived Stress | -1.19 (-1.62, -.70) | -.35 | < .001 | -.35 | -.30 | | | |
| Step 3 | | | | | | .432 | 15.73 (9, 195) | .012 |
| Sex[a] | -2.11 (-7.08, 3.20) | -.05 | .411 | -.06 | -.05 | | | |
| Age | -.18 (-.43, .06) | -.09 | .148 | -.11 | -.08 | | | |
| AF Knowledge | .09 (-.11, .28) | .05 | .396 | .06 | .05 | | | |
| Household PA | .30 (.08, .49) | .18 | .004 | .21 | .16 | | | |
| Recreational PA | -.12 (-.25, .02) | -.12 | .063 | -.14 | -.10 | | | |
| **HRQoL Symptoms subscale** | | | | | | | | |
| Step 1 | | | | | | .218 | 26.92 (2, 195) | < .001 |
| Recency of AF symptoms | 6.17 (4.54, 8.05) | .44 | < .001 | .43 | .42 | | | |
| Overall health | 2.77 (-1.59, 6.72) | .09 | .178 | .10 | .09 | | | |
| Step 2 | | | | | | .297 | 20.22 (4, 195) | < .001 |
| Overall mental health | -2.33 (-7.60, 2.41) | -.07 | .349 | -.07 | -.06 | | | |
| Perceived Stress | -1.07 (-1.53, -.58) | -.32 | < .001 | -.30 | -.27 | | | |
| Step 3 | | | | | | .357 | 11.48 (9, 195) | .005 |
| Sex[a] | -3.53 (-8.86, 1.76) | -.08 | .195 | -.10 | -.08 | | | |
| Age | .12 (-.13, .37) | .06 | .351 | .07 | .06 | | | |
| AF Knowledge | .06 (-.17, .30) | .03 | .617 | .04 | .03 | | | |
| Household PA | .35 (.13, .58) | .20 | .002 | .23 | .19 | | | |
| Recreational PA | -.22 (-.36, -.08) | -.22 | .001 | -.23 | -.19 | | | |
| **HRQoL Daily Activities subscale** | | | | | | | | |
| Step 1 | | | | | | .320 | 45.51 (2, 195) | < .001 |
| Recency of AF symptoms | 4.89 (2.65, 7.33) | .27 | < .001 | .31 | .26 | | | |
| Overall health | 17.21 (11.58, 22.36) | .43 | < .001 | .45 | .42 | | | |
| Step 2 | | | | | | .371 | 28.16 (4, 195) | < .001 |
| Overall mental health | -6.58 (-12.10, -1.04) | -.16 | .028 | -.16 | -.13 | | | |
| Perceived Stress | -1.15 (-1.75, -.55) | -.27 | < .001 | -.27 | -.22 | | | |
| Step 3 | | | | | | .436 | 15.97 (9, 195) | .001 |
| Sex[a] | -1.43 (-7.47, 4.87) | -.03 | .659 | -.03 | -.02 | | | |
| Age | -.53 (-.84, -.23) | -.20 | < .001 | -.24 | -.19 | | | |
| AF Knowledge | .11 (-.12, .35) | .05 | .389 | .06 | .05 | | | |
| Household PA | .30 (.03, .57) | .14 | .022 | .17 | .13 | | | |
| Recreational PA | -.07 (-.25, .09) | -.05 | .400 | -.06 | -.05 | | | |
| **HRQoL Treatment subscale** | | | | | | | | |
| Step 1 | | | | | | .169 | 19.57 (2, 195) | < .001 |
| Recency of AF symptoms | 5.22 (3.38, 7.04) | .36 | < .001 | .35 | .34 | | | |
| Overall health | 4.39 (-.18, 8.95) | .14 | .048 | .14 | .13 | | | |
| Step 2 | | | | | | .295 | 19.97 (4, 195) | < .001 |
| Overall mental health | -1.38 (-6.80, 4.26) | -.04 | .594 | -.04 | -.03 | | | |
| Perceived Stress | -1.34 (-1.82, -.87) | -.39 | < .001 | -.36 | -.32 | | | |

*(Continued)*

**Table 3.** (Continued)

| | B (95% CI) | β | Coefficient P value | Partial r | Semi-partial r | R Square | Model F (df) | F change P value |
|---|---|---|---|---|---|---|---|---|
| Step 3 | | | | | | .316 | 9.56 (9, 195) | < .001 |
| Sexa | -2.15 (-8.09, 4.23) | -.05 | .461 | -.06 | -.05 | | | |
| Age | .08 (-.23, .35) | .04 | .583 | .04 | .03 | | | |
| AF Knowledge | .05 (-.16, .27) | .03 | .699 | .03 | .02 | | | |
| Household PA | .25 (.01, .48) | .14 | .034 | .16 | .12 | | | |
| Recreational PA | -.11 (-.26, .05) | -.11 | .125 | -.11 | -.09 | | | |
| **AF Treatment Satisfaction** | | | | | | | | |
| Step 1 | | | | | | .202 | 24.50 (2, 195) | < .001 |
| Recency of AF symptoms | 5.99 (3.89, 8.25) | .35 | < .001 | .35 | .34 | | | |
| Overall health | 8.11 (2.71, 13.03) | .21 | .002 | .23 | .21 | | | |
| Step 2 | | | | | | .225 | 13.86 (4, 195) | .065 |
| Overall mental health | 2.31 (-3.42, 8.48) | .06 | .468 | .05 | .05 | | | |
| Perceived Stress | -.49 (-1.06, .08) | -.12 | .120 | -.11 | -.10 | | | |
| Step 3 | | | | | | .278 | 7.96 (9, 195) | .021 |
| Sexa | -6.61 (-13.71, .77) | -.12 | .062 | -.14 | -.12 | | | |
| Age | .39 (.06, .71) | .15 | .025 | .16 | .14 | | | |
| AF Knowledge | -.34 (-.61, -.09) | -.15 | .019 | -.17 | -.15 | | | |
| Household PA | .26 (-.06, .51) | .12 | .069 | .13 | .11 | | | |
| Recreational PA | -.01 (-.20, .16) | -.01 | .916 | -.01 | -.01 | | | |

Note: [a]Dummy variable, 0 = male, 1 = female; B = Unstandardized Coefficient; 95%CI = Bootstrapped 95% confidence interval; β = Standardized Beta Coefficients.
AF = Atrial fibrillation; PA = physical activity

explaining the most variance on the HRQoL daily activities subscale (32%) and the least variance in the HRQoL treatment subscale (16.9%). On Step 2 of the regressions, overall mental health was positively related to HRQoL treatment subscale and higher perceived stress was related to lower HRQoL on all three subscales, as well as lower overall HRQoL. In Step 3, age was negatively related to HRQoL for daily activities and higher household physical activity was related to higher HRQoL on all three subscales, as well as higher overall HRQoL whereas higher recreational physical activity was related to lower HRQoL for symptoms. Together the individual characteristics increased the total variance accounted for in overall HRQoL to 43.2%, accounting for an additional 13.6% of the variance over and above recency of AF symptoms. The amount of variance accounted for within each subscale was similar; notably, the highest proportion of variance these individual characteristics accounted for was in the HRQoL treatment sub-scale (14.7%), which neared (but did not pass) the contribution of AF symptoms and overall health to this outcome.

## Treatment satisfaction

On Step 1 of the regression, length of time since participants last had AF symptoms was positively related to AF treatment satisfaction. In addition, higher overall health was positively related to AF treatment satisfaction. Together, these variables accounted for 20.2% of the variance in AF treatment satisfaction. On Step 2 of the regressions, overall mental health and perceived stress did not add to the incremental prediction in AF treatment satisfaction. On step 3, age was positively related to AF treatment satisfaction, whereas greater AF knowledge was related to lower AF treatment satisfaction. Together the individual characteristics increased the total variance accounted for in overall HRQoL to 27.8%, accounting for an additional 7.6%

of the variance in AF treatment satisfaction over and above recency of AF symptoms and overall health.

## Discussion

In this study of patients attending a specialized AF clinic, we found that along with AF symptoms and overall health, individual characteristics including perceived stress and physical activity were important predictors of HRQoL and AF treatment satisfaction. These findings are consistent with those predicted by Ferrans et al.'s model of HRQoL [8].

Consistent with other well-established findings was a relationship between HRQoL and recency of AF symptoms. Indeed, in the original AFEQT scale validation study, patients who had AF symptoms within the past 4 weeks had lower mean overall AFEQT scores compared to those with more remote symptoms (57.1 vs. 66.6, P = 0.01) [21]. In the present study, overall health ratings were also associated with overall HRQoL (and the daily activities and treatment subscales), as well as AF treatment satisfaction. In a systematic review, general health perceptions fully mediated the relationship between symptoms and HRQoL among patients living with chronic illness in some studies, whereas in other studies the direct link between symptoms and HRQoL was not fully mediated by general health perceptions, consistent with our findings that both variables independently contributed to HRQoL [31]. This highlights the importance of managing and monitoring symptoms and supporting overall health for patients with AF [32].

The addition of individual characteristics to the overall HRQoL model, in particular perceived stress and physical activity (household/recreational) added significantly to the explained variance in HRQoL. These two variables were significant predictors in all three HRQoL subscales as well as in overall HRQoL scores. The additional contribution of perceived stress and physical activity in explaining HRQoL over and above recency of symptoms is an important finding. Although the individual characteristics accounted for about half the proportion of variance that recency of AF symptoms and overall health did in overall HRQoL, an additional 13.6% is still substantial variance [28]. According to Cohen's [28] benchmarks, AF symptoms and overall health were approaching having a large effect on HRQoL, whereas the individual variables had a medium effect. In addition, for the HRQoL treatment subscale, the proportion of variance accounted for by the individual characteristics (14.7%) was nearing that accounted for by AF symptoms and overall health (16.9%)—each contributing medium effects. It is not just the case that those who have more recently experienced symptoms are more stressed, but instead, stress *in itself* adds additional variance explained over and above recency of symptoms. Although prior studies have reported that patients perceive that stress triggers/causes their AF symptom episodes [10–13] or increases symptom severity [33] this is the first study to link perceived stress as a unique contributor to HRQoL in the AF population. Further, it expands on other evidence showing the impact of psychological distress, namely depression and/or anxiety on greater AF symptom severity, diminished HRQoL, and recurrence of AF [34] and draws attention to the need to address patients perceived stress in addition to symptom control in enhancing patients HRQoL and AF management [35].

The role of activity in HRQoL for patients with AF is unclear. In our study, recreational activity negatively predicted AF HRQoL symptoms; those doing higher recreational activity had lower HRQoL around symptoms, meaning their symptoms have been bothering them more. This is consistent with AF studies of endurance athletes in which moderate activity appears to reduce AF risk but intense exercise increases both incident AF [36] and AF burden; the latter, possibly due to the psychological impact of AF and medication impacts on recreational activities [37]. Interestingly, higher household physical activity was related to higher

HRQoL on all three subscales, as well as overall HRQoL. This may indicate that the ability to engage in one's daily activities for older individuals represented in our study is more important for their HRQoL than the potential benefits of recreational exercise. In fact, although regular exercise and high cardiorespiratory fitness contribute to a reduction in incident AF [36] and exercise interventions improve exercise capacity in patients with AF [38], exercise interventions do not appear to improve HRQoL in the long term [39].

Treatment satisfaction, an AFEQT component not included in the overall score, also showed significant relationships with recency of AF symptoms and overall health, similar to the subscale and overall HRQoL scores. However, age and knowledge significantly contributed over and above to explaining the variance in AF treatment satisfaction. The finding that older patients had higher satisfaction with their AF treatment is consistent with literature that suggests older adults are more satisfied with healthcare generally [40], as well as with findings that older adults (65+ years) had slightly higher AF treatment satisfaction scores both pre and post ablation compared to younger adults (<65 years) [41]. Although AF knowledge is considered to be important for promoting symptom management [17], the role of AF knowledge in HRQoL is not well defined. Possessing high knowledge may serve to raise treatment outcome expectations that when unmet, contribute to dissatisfaction.

Despite being related to AF treatment satisfaction in the present study, AF knowledge was unrelated to overall HRQoL or the three HRQoL subscales. Consistent with our findings, in a previous study targeting education to address knowledge gaps in patients with AF, in-person education improved knowledge scores but had no influence on quality of life, symptom burden or medication adherence [42]. Yet, interestingly, in another RCT, patients with AF assigned to an education intervention reported improved knowledge and higher AFEQT scores at 1- and 3- month follow-up for both overall HRQoL and treatment satisfaction [43]. More research is needed into the relationship between AF knowledge and HRQoL and AF treatment satisfaction.

Although a recent systematic review highlighted the importance of both symptoms and individual characteristics in HRQoL across diverse studies [6], our study is among the first to comprehensively co-examine both clinical and individual characteristics and the role each independently plays in HRQoL among an AF patient population. The finding that individual characteristics such as perceived stress, physical activity and AF knowledge accounted for variability in HRQoL/treatment satisfaction over and above the variability accounted for by recency of AF symptoms and overall health suggests that these variables could potentially be targets for intervention, given they are modifiable. It has long been suggested that HRQoL among patients with AF can be improved with better symptom control, and we suggest that reducing perceived stress and improving household activity could further be targeted as potential areas to improve HRQoL, although the relationship between recreational activity and AF knowledge with HRQoL may be more complicated. These results suggest the importance of considering multidimensional interventions to support HRQoL among patients with AF, and potentially considering targeting different interventions for specific groups (e.g., those high/ low in AF knowledge). Indeed, researchers have recently been observing the diversity/variability in AF populations [44], suggesting that in future we may need to move beyond assuming linearity in theorizing the relationships between personal characteristics and HRQoL.

## Strengths/limitations

This study provides an expanded understanding of the role of personal characteristics over and above AF symptoms and overall health in explaining HRQoL and contributes to further theorizing of HRQoL in the AF population. It should be noted that variables were based on

self-report, and participant responses might be subject to recall bias; however, standardized and validated measures were used, reducing potential for misinterpretation. Future research may seek to explore less subjective measures (e.g., objective assessments of physical activity, such as accelerometer readings, or objective measures of stress, such as cortisol levels). Our population was patients referred to a specialty AF clinic, and only included 7–8% who were asymptomatic; indeed patients referred to tertiary clinics often have a greater disease burden than average AF patients [6] potentially reducing the relevance of our findings to general AF populations, where closer to 25 or 30% of patients are typically asymptomatic populations [1]. In an effort to reduce sampling bias, all patients with upcoming appointments were invited to participate. Because 34% of eligible participants completed the survey, responses may be influenced by self-selection bias; further, patient participants may not be representative of the diversity of the AF population as they are predominantly well-educated, high income, male, and Caucasian; however, they reflect the larger AF clinic population (i.e., the clinic participants were sampled from had a similar demographic breakdown). Nevertheless, overall, our selection of patients referred to a specialty clinic likely resulted in a sample with more high risk/advanced AF patients (e.g., many referred for ablation), limiting our ability to generalize the findings to broader populations of AF patients (e.g., those under primary care). Future research is needed to determine whether psychological, demographic, and behavioral characteristics play a larger or smaller role in HRQoL relative to AF symptoms and overall health for broader populations of patients with AF.

## Conclusion

Patients with AF receiving care at an AF specialty clinic had moderate HRQoL and treatment satisfaction, with higher HRQoL among patients who had not experienced AF symptoms in over a month. Findings showed that individual characteristics, namely perceived stress and physical activity made unique contributions in explaining HRQoL over and above AF symptoms and overall health. Similarly, the individual characteristics age and AF knowledge contributed additionally to AF symptoms and overall health in explaining the variance in AF treatment satisfaction. There has been considerable emphasis on symptoms in predicting HRQoL in the AF population. These findings point to the need to give increasing attention to other characteristics of the individual as they play an important role in AF patients' HRQoL.

## Acknowledgments

The authors wish to extend their thanks to all the patient participants who shared their insights during the study and to Kaylee Neill and Sarah Singh, who assisted with recruitment.

## Author Contributions

**Conceptualization:** Kathy L. Rush, Lindsay Burton, Peter Loewen, Brian P. O'Connor, Jason G. Andrade.

**Data curation:** Kathy L. Rush, Kendra Corman.

**Formal analysis:** Kathy L. Rush, Cherisse L. Seaton, Kendra Corman.

**Funding acquisition:** Kathy L. Rush.

**Investigation:** Kathy L. Rush, Lana Moroz, Mindy A. Smith, Jason G. Andrade.

**Methodology:** Kathy L. Rush, Lindsay Burton.

**Project administration:** Kathy L. Rush, Lindsay Burton.

**Resources:** Lana Moroz, Jason G. Andrade.

**Supervision:** Kathy L. Rush, Cherisse L. Seaton, Lindsay Burton, Brian P. O'Connor.

**Visualization:** Cherisse L. Seaton.

**Writing – original draft:** Kathy L. Rush, Cherisse L. Seaton, Lindsay Burton, Mindy A. Smith.

**Writing – review & editing:** Kathy L. Rush, Cherisse L. Seaton, Lindsay Burton, Peter Loewen, Brian P. O'Connor, Lana Moroz, Kendra Corman, Mindy A. Smith, Jason G. Andrade.

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
