## [Decision Letter · Decision Letter 0]

19 Jun 2023

PONE-D-23-08648Quality of life among patients with atrial fibrillation:  A Theoretically-guided Cross-Sectional StudyPLOS ONE

Dear Dr. Rush,

Thank you for submitting your manuscript to PLOS ONE. After careful consideration, we feel that it has merit but does not fully meet PLOS ONE’s publication criteria as it currently stands. Therefore, we invite you to submit a revised version of the manuscript that addresses the points raised during the review process.Article language especially in the introduction of the manuscript is very complicated and difficult to understand on the first read. Please note comments by reviewers. I highlighted the top 2 as noted below Variance change should be explained in more simple terms for readers to understand the significance of the study Please refer to the reviewer comments suggesting making the study more narrative without dividing hypothesis by number and to assess for selection bias.Please refer to other reviewer comments and have a response.

We look forward to receiving your revised manuscript.

Kind regards,

Vikramaditya Samala Venkata

Academic Editor

PLOS ONE

Journal Requirements:

This work was supported by a Canadian Institutes of Health Research Project Grant (award number PJT-148737; principal investigator, KLR).

However, funding information should not appear in the Acknowledgments section or other areas of your manuscript. We will only publish funding information present in the Funding Statement section of the online submission form. 

This work was supported by a Canadian Institutes of Health Research (https://cihr-irsc.gc.ca) Project Grant, award number PJT-148737; principal investigator, KLR. The funders had no role in study design, data collection and analysis, decision to publish, or preparation of the manuscript.

I have read the journal's policy and the authors of this manuscript have the following competing interests: JGA reports grants from Medtronic and the Heart and Stroke Foundation of Canada during the conduct of this study; personal fees from Medtronic and Biosense Webster Inc, outside the submitted study. The other authors have no conflicts to declare. 

Reviewers' comments:

Reviewer's Responses to Questions

**Comments to the Author**

1. Is the manuscript technically sound, and do the data support the conclusions?

Reviewer #1: Yes

Reviewer #2: Yes

Reviewer #3: Yes

Reviewer #4: Yes

Reviewer #5: Yes

Reviewer #6: Yes

2. Has the statistical analysis been performed appropriately and rigorously? 

Reviewer #1: Yes

Reviewer #2: I Don't Know

Reviewer #3: Yes

Reviewer #4: Yes

Reviewer #5: Yes

Reviewer #6: Yes

3. Have the authors made all data underlying the findings in their manuscript fully available?

Reviewer #1: Yes

Reviewer #2: Yes

Reviewer #3: Yes

Reviewer #4: Yes

Reviewer #5: Yes

Reviewer #6: Yes

4. Is the manuscript presented in an intelligible fashion and written in standard English?

Reviewer #1: Yes

Reviewer #2: Yes

Reviewer #3: Yes

Reviewer #4: Yes

Reviewer #5: Yes

Reviewer #6: Yes

5. Review Comments to the Author

Reviewer #1: Overall a well designed study with a good analysis of results with a reasonable conclusion. Not sure why Hypothesis 1 was included in the study as the Authors mention in the introduction that studies have already been conducted on it and the Hypothesis has already been proven. Other than that, well executed study.

Reviewer #2: Well presented original cross sectional research article. Atrial fibrillation patients with individual characteristics including perceived stress, household physical activity, AF treatment satisfaction, AF knowledge, recent AF symptoms <1 month being important overall predictors of Health related quality of life are well described. Data acquirement is well explained.

Reviewer #3: Appreciate the effort for this well presented abstract. The physician community is highly focused on symptoms rather the individual factors considered here. This paper should help bring more attention to the less appreciated characteristics.

Reviewer #4: Quality of life with atrial fibrillation is well picked topic. However I see a lot of confounders as the classification of atrial fibrillation is not taken into account. Paroxysmal atrial fibrillation has a different set of symptoms or a different frequency of symptoms and Persistent and permanent atrial fibrillation have various set of symptoms as well. A lot of these patients have multiple comorbidities and various patients have a different regimen for treatment. It would have been great if there is more granularity in choice of patients and cohorts. Also a lot of patients may have had procedures like ablation and pacemaker implantations. Some have heart failure as a comorbidity which can worsen the quality of life. Statistically may be sound but the as mentioned above choice of patients should have been tighter and not just every patient that comes to an A.fib clinic. Moreover it is a very subjective study.

Reviewer #5: This is a study conducted on a population of Atrial fibrillation(AF) patients attending a specialty clinic. Authors have used hierarchal regression analyses to assess whether individual characteristics of patients are predictors of Health-related Quality of Life(HRQoL) and AF treatment satisfaction. Prior studies have explored these contributions with more focus on AF symptoms rather than individual characteristics. The findings are similar to other studies that authors have cited.

The authors have given enough background to justify the need for this study. The stated objectives of the study are clear. Results have been reported in a satisfactory manner. They have outlined the strength and limitations of their study in detail. They have reported reliability and validity for all the instruments used in the study.

This study tested unexplored individual characteristics including perceived stress and AF knowledge. It has been investigated before whether Perceived stress contributes to mental health and anxiety in AF patients (DA Lane et al, 2009) but it has not been investigated for contribution to HRQoL.

Authors have claimed that physical activity has not been studied as an individual characteristic which is not true. Exercise performance has been found to have a significant contribution to AF-related HRQoL by SN Singh et al in 2006. In this study, authors have further dichotomized physical activity into household activity and recreational activity which has given contrasting and interesting results.

Authors have reported taking help from family members if the patient could not understand English. They should elaborate on why a certified translator was not used if language was a barrier.

Authors should explain the concept of variance change in more simple terms to readers who need to become more familiar with the statistical significance of this measure.

Reference 9 is incomplete. It should be completed.

Lane D.A., Langman C.M., Lip G.Y., Nouwen A. Illness perceptions, affective response, and health-related quality of life in patients with atrial fibrillation. J. Psychosom. Res. 2009;66:203–210.

Singh S.N., Tang X.C., Singh B.N., Dorian P., Reda D.J., Harris C.L., Fletcher R.D., Sharma S.C., Atwood J.E., Jacobson A.K. Quality of life and exercise performance in patients in sinus rhythm versus persistent atrial fibrillation: A Veterans Affairs Cooperative Studies Program Substudy. J. Am. Coll. Cardiol. 2006;48:721–730. doi: 10.1016/j.jacc.2006.03.051.

Reviewer #6: the present is an interesting single center study.

Some issues should be addressed.

introduction: maybe it may be worth to make it more narrative, without dividing hyphothesis by number

methods: the referral to a specialized AF clinics may hide a selection bias towards high risk patients or for those referred to af ablation. please comment

methods: the statistical analysis is well performed and fit from a clinical point of view. Maybe (from a formal point of view) also performing a multivariate analysis with all the predictors may be worth of.

6. PLOS authors have the option to publish the peer review history of their article (what does this mean?). If published, this will include your full peer review and any attached files.

Reviewer #1: No

Reviewer #2: **Yes: **Jyotsna Gummadi MD

Reviewer #3: No

Reviewer #4: **Yes: **PARITHARSH GHANTASALA

Reviewer #5: No

Reviewer #6: **Yes: **Fabrizio D'Ascenzo

---

## [Author Response · Author response to Decision Letter 0]

4 Aug 2023

Editor Comments 

1. Article language especially in the introduction of the manuscript is very complicated and difficult to understand on the first read. Please note comments by reviewers. I highlighted the top 2 as noted below Thank you, we have simplified the language, as suggested.

2. Variance change should be explained in more simple terms for readers to understand the significance of the study We have added a simplified explanation of % change, and a reference to Cohen’s benchmarks in the data analysis section to aid interpretation of these.

3. Please refer to the reviewer comments suggesting making the study more narrative without dividing hypothesis by number and to assess for selection bias. We have made the study more narrative without dividing the hypotheses by number, and we have added greater clarification to the limitations section re: selection bias.

4 Please refer to other reviewer comments and have a response. Completed as requested (see below).

 Reviewer 1 Comments 

 Overall a well designed study with a good analysis of results with a reasonable conclusion. Not sure why Hypothesis 1 was included in the study as the Authors mention in the introduction that studies have already been conducted on it and the Hypothesis has already been proven. Other than that, well executed study. Thank you for this feedback. It is correct that studies have already been conducted on hypothesis 1 (i.e., the link between patient symptoms and QoL); however this was included primarily so that we could explore whether the other variables were significantly related to QoL outside of symptoms. If we hadn’t entered symptoms and overall health in the regression models first, we would not be able to determine if variables such as exercises were independently related to QoL, or whether both exercise and QoL were simply varying together because of symptoms. We have simplified language about variance change in order to try and make this clearer, and we have removed reference to individual hypotheses, as per other reviewer suggestions, which re-directs focus to the key hypotheses.

 Reviewer 2 Comments 

 Well presented original cross sectional research article. Atrial fibrillation patients with individual characteristics including perceived stress, household physical activity, AF treatment satisfaction, AF knowledge, recent AF symptoms <1 month being important overall predictors of Health related quality of life are well described. Data acquirement is well explained. Thank you for this feedback.

 Reviewer 3 Comments 

 Appreciate the effort for this well presented abstract. The physician community is highly focused on symptoms rather the individual factors considered here. This paper should help bring more attention to the less appreciated characteristics. Thank you for this feedback.

 Reviewer 4 Comments 

 Quality of life with atrial fibrillation is well picked topic. However I see a lot of confounders as the classification of atrial fibrillation is not taken into account. Paroxysmal atrial fibrillation has a different set of symptoms or a different frequency of symptoms and Persistent and permanent atrial fibrillation have various set of symptoms as well. A lot of these patients have multiple comorbidities and various patients have a different regimen for treatment. It would have been great if there is more granularity in choice of patients and cohorts. Also a lot of patients may have had procedures like ablation and pacemaker implantations. Some have heart failure as a comorbidity which can worsen the quality of life. Statistically may be sound but the as mentioned above choice of patients should have been tighter and not just every patient that comes to an A.fib clinic. Moreover it is a very subjective study. Thank you for this clear description with respect to how patient populations with AF may vary.

Clinic referral data abstraction was completed for these patients as part of another study. As such, we do know clinical details for this sample as a whole and have added a summary to the results section. 

Indeed, our patient population included diversity with respect to AF classification, and this reviewer is correct that many were referred, at least in part, for ablation consultation (as 43% of clinicians mentioned ‘ablation consultation’ as part of the referral), and many had already undergone previous ablations, had co-morbidities, etc. 

We have added a brief summary description of these patient clinic characteristics as well as expanded on the implications of this selection bias in the limitations.

Tighter selection of patients (e.g., focused just on those with advanced/persistent AF and heart failure) would have allowed for a very granular look at a sub-population, but may not have allowed for generalizability across more diverse sub-populations. The diversity within our sample was a strength because we were able to demonstrate that even when AF symptoms and overall health were considered (i.e., “controlled” in the regressions), AF patients’ psychological, demographic, and behavioral characteristics were associated with HRQoL. Whether this finding holds true for all of the different patients represented in our sample is a good direction for future research. 

We appreciate the view that this is a very subjective study, as that is the nature of many self-report inventories, and we have added this to the limitations as a suggestion for future research.

 Reviewer 5 Comments 

 This is a study conducted on a population of Atrial fibrillation(AF) patients attending a specialty clinic. Authors have used hierarchal regression analyses to assess whether individual characteristics of patients are predictors of Health-related Quality of Life(HRQoL) and AF treatment satisfaction. Prior studies have explored these contributions with more focus on AF symptoms rather than individual characteristics. The findings are similar to other studies that authors have cited. Thank you for this feedback.

 The authors have given enough background to justify the need for this study. The stated objectives of the study are clear. Results have been reported in a satisfactory manner. They have outlined the strength and limitations of their study in detail. They have reported reliability and validity for all the instruments used in the study.

 Thank you for this feedback.

 This study tested unexplored individual characteristics including perceived stress and AF knowledge. It has been investigated before whether Perceived stress contributes to mental health and anxiety in AF patients (DA Lane et al, 2009) but it has not been investigated for contribution to HRQoL. Thank you for this feedback. We have added information about the link between perceived stress and mental health/anxiety.

 Authors have claimed that physical activity has not been studied as an individual characteristic which is not true. Exercise performance has been found to have a significant contribution to AF-related HRQoL by SN Singh et al in 2006. In this study, authors have further dichotomized physical activity into household activity and recreational activity which has given contrasting and interesting results.

 Thank you for drawing our attention to this work. We have modified wording to say that exercise has been ‘under-explored’ (as opposed to ‘unexplored’), and have added a description of and reference to Singh et al. (2006).

 Authors have reported taking help from family members if the patient could not understand English. They should elaborate on why a certified translator was not used if language was a barrier.

 Language was not a barrier as patients attending the clinic and comprising our sample were primarily white, highly educated, and spoke English. Instead, the suggestion to have a family member assist was due to concern that elderly patients may not be able to navigate the online survey alone. We have modified the wording for clarity.

 Authors should explain the concept of variance change in more simple terms to readers who need to become more familiar with the statistical significance of this measure.

 We have added additional explanation of % variance, including a reference to Cohen for interpretation of small, medium and large effects based on % change.

 Reference 9 is incomplete. It should be completed.

 Thank you for noting this. We have corrected the Ferrans et al reference (now reference 8).

 Reviewer 6 Comments 

 the present is an interesting single center study.

Some issues should be addressed. Thank you for this feedback.

 introduction: maybe it may be worth to make it more narrative, without dividing hyphothesis by number Thank you for this suggestion, we have edited as suggested.

 methods: the referral to a specialized AF clinics may hide a selection bias towards high risk patients or for those referred to af ablation. please comment This is plausible that our focus on patients referred to a specialty clinic likely resulted in selection of a higher risk/more advanced AF population, as our AF symptom data indicates. We have enhanced the description of this and the implications of it in the limitations section.

 methods: the statistical analysis is well performed and fit from a clinical point of view. Maybe (from a formal point of view) also performing a multivariate analysis with all the predictors may be worth of. We are unsure what type of multivariate analysis the reviewer is referring to here. Multivariate analysis (MVA) includes any analyses evaluating multiple variables (more than two) to identify any possible associations. We chose linear regression as it allowed for the examination of associations between continuous variables; we also provide a table of correlations between all variables (Table 2), and we conducted a multidimensional scaling analysis with all predictors entered simultaneously to confirm which variables were most closely related to HRQoL. We have now highlighted this multidimensional scaling analysis further in the text for clarity.

---

## [Decision Letter · Decision Letter 1]

1 Sep 2023

Quality of life among patients with atrial fibrillation:  A theoretically-guided cross-sectional study

PONE-D-23-08648R1

Dear Dr. Rush,

We’re pleased to inform you that your manuscript has been judged scientifically suitable for publication and will be formally accepted for publication once it meets all outstanding technical requirements.

Kind regards,

Vikramaditya Samala Venkata

Academic Editor

PLOS ONE

Additional Editor Comments (optional):

Reviewers' comments:

Reviewer's Responses to Questions

**Comments to the Author**

1. If the authors have adequately addressed your comments raised in a previous round of review and you feel that this manuscript is now acceptable for publication, you may indicate that here to bypass the “Comments to the Author” section, enter your conflict of interest statement in the “Confidential to Editor” section, and submit your "Accept" recommendation.

Reviewer #5: All comments have been addressed

2. Is the manuscript technically sound, and do the data support the conclusions?

Reviewer #5: Yes

3. Has the statistical analysis been performed appropriately and rigorously? 

Reviewer #5: Yes

4. Have the authors made all data underlying the findings in their manuscript fully available?

Reviewer #5: Yes

5. Is the manuscript presented in an intelligible fashion and written in standard English?

Reviewer #5: Yes

6. Review Comments to the Author

Reviewer #5: Authors have addressed all the raised questions and concerns. They have made appropriate changes to the manuscript to make it more narrative, explained variance better, answered queries raised by all the reviewers and changed the references.

7. PLOS authors have the option to publish the peer review history of their article (what does this mean?). If published, this will include your full peer review and any attached files.

Reviewer #5: No

---

## [Editor Report · Acceptance letter]

28 Sep 2023

PONE-D-23-08648R1 

Quality of life among patients with atrial fibrillation:  A theoretically-guided cross-sectional study 

Dear Dr. Rush:

I'm pleased to inform you that your manuscript has been deemed suitable for publication in PLOS ONE. Congratulations! Your manuscript is now with our production department. 

Kind regards, 

on behalf of

Dr. Vikramaditya Samala Venkata 

Academic Editor

PLOS ONE